# Phylogenetic Analysis Based on DNA Barcoding and Genetic Diversity Assessment of *Morinda officinalis* How in Vietnam Inferred by Microsatellites

**DOI:** 10.3390/genes13111938

**Published:** 2022-10-25

**Authors:** Thanh Pham, Quynh Thi Nguyen, Duc Minh Tran, Hoi Nguyen, Hung Thai Le, Que Thi Hong Hoang, Yen Thi Van, Thang Nam Tran

**Affiliations:** 1Department of Biology, University of Education, Hue University, 34 Le Loi, Hue 530000, Vietnam; 2Faculty of Forestry, University of Agriculture and Forestry, Hue University, 102 Phung Hung, Hue 530000, Vietnam

**Keywords:** conservation, genetic variability, *M. officinalis*, population structure

## Abstract

*Morinda officinalis* How is well-known as a valuable medicinal plant found in some regions of Vietnam. This species is mainly used for treating male impotence, irregular menstruation, and rheumatoid arthritis. This study aimed to identify the species of and genetic diversity in three *M. officinalis* populations: one each in Quang Binh (QB), Thua Thien Hue (TTH), and Quang Nam (QN). In this study, four DNA barcoding markers (*ITS1*, *ITS2*, *matK*, and *rbcL*) were used to identify the species and 22 microsatellite markers were applied for population structure and diversity analyses. The results showed that the sequences of gene regions studied in *M. officinalis* had a high similarity (>95%) to the *ITS1*, *ITS2*, *matK*, and *rbcL* sequences of *M. officinalis* on BLAST. Of the four DNA barcoding markers used, *ITS1* and *ITS2* showed higher efficiency in DNA amplification of *M. officinalis*. From this study, 27 GenBank codes were published on BLAST. The results also revealed high levels of genetic diversity in populations. The average observed and expected heterozygosity values were H_O_ = 0.513 and H_E_ = 0.612, respectively. The average F_ST_ value was 0.206. Analysis of molecular variance (AMOVA) showed 70% variation within populations and 30% among populations. The population structure of *M. officinalis* inferred in STRUCTURE revealed that the optimum number of genetic groups for the admixture model was K = 2. These findings provided vital background information for future studies in the conservation of *M. officinalis* in both ex situ and in situ plans.

## 1. Introduction

*Morinda officinalis* is a perennial vine mainly distributed in tropical and subtropical regions [1]. In Vietnam, *M. officinalis* is found in the wild in provinces such as Cao Bang, Lao Cai, Ha Giang, Quang Binh, Thua Thien Hue, and Quang Tri [2,3,4].

*M. officinalis* contains various bioactive components and has been used for decades as a tonic and an antirheumatic medicinal herb in some Asian countries [2,5,6]. The root of *M. officinalis* has long been used as a tonic or nutrient supplement for alleviating diseases such as depression, Alzheimer’s disease, impotence, osteoporosis, and rheumatoid arthritis [6].

In Vietnam, due to the rapid increase in the demand for medicinal herbs, this plant has been over-exploited, leading to the depletion of raw materials. Additionally, because *M. officinalis* has slow growth and poor regeneration, its natural population has significantly shrunk and become endangered [2,4]. With the natural population reduced by at least 50%, this species has recently been classified as an endangered precious medicinal plant species [7]. Therefore, it is necessary to study the genetic diversity and structure of the natural population in order to conserve it effectively.

Nowadays, phylogenetic studies are necessary to conserve rare medicinal plants. In addition to representing the relationships among species in the tree of life, phylogenetic studies provide a framework for interdisciplinary investigations in taxonomy, evolutionary biology, biogeography, ecology, and conservation [8]. More recently, phylogenetic approaches based on molecular data have also proven to be an indispensable tool for genome comparisons. These approaches are used to identify genes and regulatory elements, interpret modern and ancient individual genomes, and reconstruct ancestral genomes [9], providing conservationists with background information to make conservation policies efficiently.

DNA barcoding is a universally used and reliable method of identifying plant species and has become a major focus in the fields of biodiversity and conservation. This molecular technique is not influenced by external factors or development stage, and DNA can be easily isolated from all tissues, providing an important basis for species identification at the genetic level [10]. Recently, *rbcL* and *matK* plastid coding genes were recommended as barcodes for plant species and have become the most used markers in flowering plants, as the rbcL region is highly suitable for amplification and sequencing [10,11]. Meanwhile, the nuclear *ITS* region includes both the *ITS1* and *ITS2* regions, with relatively strong discrimination power, that serve as complementary barcodes to matK and rbcL in plants. The four markers (*ITS1*, *ITS2*, *MatK*, and *rbcL*) used in this study have also proven effective in identifying medicinal species [12,13,14].

Currently, molecular markers help detect variations or polymorphisms that exist among individuals in the population for specific regions of DNA [15]. Among commonly used molecular markers such as AFLP (amplified fragment length polymorphism), RAPD (random amplified polymorphic), SSR (simple sequence repeat), and ISSR (inter simple sequence repeat), simple sequence repeat (SSR) markers are useful tools for research in plant genetics, breeding, and identification of individuals and species due to the allelic sequence diversity. SSRs are widely spread in the genome and have high codominant inheritance, polymorphism, and multiallelic variation [16,17,18,19].

Studies on molecular markers have been carried out on *M. officinalis* [5,20]. Liao et al. in particular developed an SSR marker dataset to serve in further research related to this plant.

The SSR marker has the advantage of using only a tiny amount of DNA. In addition, this method involves a more straightforward, faster, and more cost-effective technique than other methods.

In this study, before evaluating the genetic diversity and population structure of *M. officinalis* by using SSR markers, we used DNA barcoding to identify this species in three different provinces in central Vietnam (Quang Binh, Thua Thien Hue, and Quang Nam). The current study aimed to pave the way for protecting wild *M. officinalis* populations.

## 2. Materials and Methods

### 2.1. Plant Materials

The leaves of 37 *M. officinalis* trees were randomly collected from 3 populations in the central provinces of Quang Binh, Thua Thien Hue, and Quang Nam in Vietnam. In the field, the samples were placed in plastic bags containing silica gel. Next, the samples were transferred to the Genetic Laboratory of the Biology Department at Hue University of Education and stored at −20 °C until DNA extraction. Sampling locations were recorded using a global positioning system (GPS) (Figure 1, Appendix A).

### 2.2. DNA Extraction

Genomic DNA was extracted from 100 mg of young leaf samples using a Plant Genomic DNA Extraction Kit (Bioteke Corporation, Beijing, China) according to the manufacturer’s protocol. The quality of the extracted DNA was further estimated via 0.8% agarose gel electrophoresis. Safe dye (Phu Sa Biochem, Can Tho, Vietnam) was applied for DNA gel stain. Gel imaging was performed using a UV gel imaging system (Major Science, Saratoga, CA, USA). The isolated DNA was then stored at −20 °C until further analysis.

### 2.3. DNA Barcoding Amplification and Sequencing

A polymerase chain reaction (PCR) was performed using standard universal plant DNA barcoding primers (Table 1), with 25 μL of the reaction mixture containing 2.0 μL of template DNA (50 ng), 12.5 µL of 2X Taq Master Mix, 0.5 µL of each primer (10 pmol) (Table 2), and 9.5 µL of deionized water. PCR amplification was carried out with an Aeris™ PCR Aeris Thermal Cycler (21 Changi South Street 1, Singapore). The PCR conditions followed those laid out in the previous publication [8].

To check the presence or absence of bands, amplified PCR products were electrophoresed using 0.8% agarose gel (1 × TAE buffer and 5 μL/mL safe dye). Gel imaging was carried out using a UV gel imaging system (Major Science, Saratoga, CA, USA). Band size of amplified products was determined using a Thermo Scientific GeneRuler 100 bp DNA Ladder (Thermo Fisher Scientific, Waltham, MA, USA). 

The PCR products were sent to First Base Laboratories Sdn. Bhd (Taman Serdang Perdana, Seri Kembangan, Selangor, Malaysia) for purification and sequencing service, using the same primers as those used for the PCR.

### 2.4. Microsatellite Amplification

In all, 37 genomic DNA samples were used in this study. A polymerase chain reaction (PCR) was performed in 25 μL of reaction mixture that contained 2.0 μL of template DNA, 12.5 µL of 2X Taq Master Mix, 0.5 µL of each primer, and 9.5 µL of deionized water.

Liao et al. [20] have described the 22 microsatellite loci used to generate data for the current study (Appendix A).

PCR amplification was carried out with an Aeris™ PCR Aeris Thermal Cycler as follows: Initial denaturation was carried out at 95 °C for 3 min. This was followed by 35 cycles of 45 s each at 94 °C for denaturation, 45 s of alignment at the annealing temperature (50–52 °C) for each primer pair, and 45 s of alignment at 72 °C for extension. Finally, 10 min of alignment at 72 °C for the final cycle completed the extension of any remaining products. The samples were kept at 4 °C until they were analyzed.

The amplification products were separated on 8% polyacrylamide gels in 1 × TAE buffer using the Mini Vertical Gel Electrophoresis Apparatus (Major Science, Saratoga, CA, USA) and visualized with safe dye (Phu Sa company). A UV gel imaging system (Major Science, Saratoga, CA, USA) was used to verify the presence of amplified fragments.

### 2.5. Data Analysis

#### 2.5.1. DNA Barcoding

To obtain the sequence of each region (*ITS1*, *ITS2*, *matK*, and *rbcL*), the forward and reverse sequences were aligned using BioEdit version 7.2.5 software [21]. In searching for the similarities between those sequences and the sequences deposited in the GenBank database, the sequences of this study were analyzed using the BLAST (Basic Local Alignment Search Tool) program at http://www.ncbi.nlm.nih.gov/BLAST (accessed on 20 June 2022) [22] (Appendix A).

The alignment was then exported to Molecular Evolutionary Genetics Analysis (MEGA-X) software for phylogenetic analysis [23]. The maximum likelihood trees were constructed for *ITS1*, *ITS2*, *matK*, and *rbcL* data separately using the Kimura 2-parameter model [24] with 1000 bootstrap replicates for node supports.

#### 2.5.2. Genetic Diversity

Size of bands was detected by GelAnalyzer 19.1 software (www.gelanalyzer.com (accessed on 4 January 2022), by Istvan Lazar Jr. and Istvan Lazar Sr).

To determine the level of genetic variation within a population, the following genetic diversity parameters were calculated: mean number of alleles per locus (A), number of unique alleles, the effective number of alleles (Ne), average observed heterozygosity (Ho), average expected heterozygosity (He), and fixation index (F_IS_). All calculations were performed in GenAIEx v.6.5 [25].

A genetic distance matrix of pairwise F_ST_ values was also used to perform a hierarchical analysis of molecular variance (AMOVA) in GenAIEx v.6.5 [25]. Significance levels were determined using 999 permutations. AMOVA was used to estimate and partition the total variances at two hierarchy levels: within populations and among populations.

Principal coordinate analysis (PCoA) based on the codominant genotypic distance for 37 studied samples of 3 populations of *M. officinalis* was carried out in GenAIEx [25].

To determine the optimal value of the genetic clusters (K), a Bayesian analysis of the population structure was performed with STRUCTURE v.2.3.4 (https://web.stanford.edu/group/pritchardlab/structure.html (accessed on 28 July 2022) [26]. Once the admixture model was set with a correlated allele frequency and ancestry models, 10 separate runs of the number of clusters (K) in the dataset were performed from 1 to 10 for each K value at 500,000 Markov Chain Monte Carlo (MCMC) repetitions and a 100,000 burn-in period. The optimal value of K was detected using Structure Harvester [27] based on the ΔK value by Evanno et al. [28].

## 3. Results

### 3.1. DNA Barcoding

For this study’s DNA barcoding amplification process, we selected three samples per population, using four different primers to identify species. The results showed that *ITS1* and *ITS2* markers gave fragments of molecular weight as expected and bands of PCR products were clear for further research in all samples of the three populations, whereas some samples that used *matK* and *rbcL* markers for amplification had negative results and therefore their sequences were not determined.

A total of 27 assembled sequences were obtained and deposited in the GenBank: nine assembled sequences of *ITS1* amplicon, nine assembled sequences of *ITS2* amplicon, four assembled sequences of *matK* amplicon, and five assembled sequences of *rbcL* amplicon. The BLAST tool was used to identify the plant for each sequence, and the closest species in the GenBank was obtained. The results showed that all markers used and plants deposited in this study belonged to the genus *Morinda*. *Gynochthodes officinalis* (synonym of *M. officinalis*) was the most closely related species, with a higher extent than other species of the *Morinda* genus.

In our study, *ITS1* and *ITS2* were the most effective in terms of the amplification process (100%) (Table 2).

In the case of ITS1 region sequences (~179 bp), through use of the *ITS1* pair of primers (*ITS 5a* forward and 4 reverse) and the BLAST tool for identification, the closest species was found to be *M. officinalis*, at 98.88% similarity (Table 3). The phylogenetic tree for those sequences was constructed with the closest species (Figure 2). In the case of the *ITS2* region sequences (~235 bp), through use of the *ITS2* pair of primers (*ITS S2F* and *ITS S3R*), the closest species was found to be *Gynochthodes officinalis* (synonym of *M. officinalis*), at 100% similarity (Table 4). The phylogenetic tree for those sequences was constructed with the closest species (Figure 3).

In the case of the *matK* region sequences (ranging from 861 bp to 899 bp), through use of the *matK* pair of primers (*3**90F* and *1326R*), the closest species was found to be *M. officinalis*, at 99.89% similarity (Table 5), followed by *Gynochthodes parvifolia*, at 99.78% similarity. The phylogenetic tree for these sequences was constructed with the closest species (Figure 4).

In use of the *rbcL* gene, the BLAST result on NCBI showed that the nucleotide sequences obtained were highly similar to those of species of *Morinda officinalis* (accession numbers KR869730.1, NC_053818.1, and GQ436556.1), *G. parvifolia* (accession number NC_054151.1), and *Morinda* sp. SH-2010 (accession numbers AB586541.1 and AB586543.1), with similarity ranging from 99.31% to 100% (Table 6). The phylogenetic tree for those sequences was constructed with the closest species (Figure 5).

### 3.2. Nucleotide Components

In terms of the occurrence of each type of nucleotide in the *ITS1* and *ITS2* gene region, cysteine (C) accounted for the highest proportion, ranging from 35.3% to 38.9%, followed by guanidine (G), which ranged from 31.1% to 34.0%. Timin (uracil) was observed with the lowest percentage (ranging from 10.0% to 14.5%) (Table 7). For matK and rbcL, however, the percentage of Timin (uracil) was higher than that of the remaining nucleotides (ranging from 28.1% to 36.9%) (Table 8).

The (G + C) percentage was the highest, with an average of 69.6% in the *ITS2* gene region (Table 7 and Table 8).

### 3.3. Genetic Diversity and Population Structure

A total of 140 different alleles were detected from 22 microsatellite markers in *M. officinalis* in the three surveyed populations, with sizes ranging from 106 bp to 328 bp.

In this study, of all 22 loci, the locus MO05 showed the highest number of alleles in all populations (Na = 9), while the locus MO90 showed the lowest number of alleles, with Na = 2.333. The F_IS_ values of the loci ranged between −0.287 and 0.965, with an average of 0.216. Half of the loci (MO02, MO04, MO05, MO12, MO26, MO30, M039, MO41, MO60, MO63, and MO96) showed positive F_IS_ values (Table 9). The F_IT_ values of the loci ranged between −0.086 and 0.978.

All the loci showed positive F_ST_ values ranging between 0.127 and 0.554, with an average of 0.262. This result indicated a relatively high genetic differentiation at the locus level in all populations (Table 9).

Table 10 presents the genetic diversity of *M. officinalis* populations. The allele numbers (Na) ranged from 2 to 8 in the QB population, from 3 to 12 in the TTH population, and from 0 to 10 in the QN population. Specifically, the TTH population had the highest number of alleles, with an average of 6.455, whereas the population with the lowest number of alleles (3.955) was the QB population (Table 10). The average percentage of polymorphic loci was 98.48%, of which two populations, QB and TTH, reached the absolute value (100%).

The highest total number of private alleles (Np = 4.364) was observed in the TTH population, followed by the QN population (Np = 2.318) and the QB population (Np = 1.818). Similarly, of the three populations, the effective allele numbers of the TTH population were the highest at 3.786, compared to 3.221 and 2.567 for the QN and QB populations, respectively.

Observed heterozygosity (Ho) and expected heterozygosity (He) averaged 0.513 and 0.612, respectively. The QN population had the highest expected heterozygosity (0.591) of the three populations, whereas the value of the observed heterozygosity of that population was 0.601. The fixation index (F_ST_) was positive for all populations (F_ST_ = 0.206).

The results of the AMOVA pointed to higher levels of variation within populations (70%) than between populations (30%) (Table 11), indicating a relatively low genetic divergence among the three studied populations (TTH, QB, QN). However, within populations, individuals were found to have high genetic differentiation.

An admixture model was performed to determine the group population of the 37 individuals of *M. officinalis* based on Bayesian analysis using the STRUCTURE program. The population structure of *M. officinalis* inferred in STRUCTURE revealed that the optimum number of genetic groups for the admixture model was K = 2, with a delta K value of 74.084117. After a PCA analysis, the three populations were classified into two distinct clusters. K = 2, thus, can be considered to be the most suitable number of clusters for the structure of the populations in this study (Figure 6 and Figure 7). The QB and QN populations were in one cluster (green cluster), whereas the TTH population was in the remaining cluster (red cluster) (Figure 7).

## 4. Discussion

DNA barcoding is a novel approach for identifying and classifying species based on the nucleotide diversity of conserved sequences. Recently, many studies have indicated that DNA markers *ITS1*, *ITS2*, *matK*, and *rbcL* are highly effective in identifying medicinal plants at species and genus levels. Therefore, this study was conducted to investigate the efficiency of those markers in barcoding *M. officinalis* plant species for the first time to identify the best marker for this valuable plant.

In previous studies, the *matK* gene and (*ITS*) region genetic markers have been proven highly efficient in distinguishing plant species and therefore identified as potential candidates for barcoding plants [29]. This study demonstrated that the *ITS1* and *ITS2* regions of *M. officinalis* were amplified more effectively compared to the other two markers (*matK* and *rbcL*).

Sequence analysis using the above four primers showed that the samples studied in this experiment belonged to the genus *Morinda* and the most closely related species was *Gynochthodes officinalis* (synonym of *M. officinalis)*. In the present study, the *ITS2* marker showed the highest efficiency in terms of amplification and species identification. The higher capacity of *ITS2* in terms of phylogenetic reconstruction has been proven in previous studies [12,13,29].

The origin and ecology of a species are often expressed through its genetic diversity [30]. Many previous studies have proven a positive correlation between population genetic diversity, population size, and geographic distribution range [30,31]. Species with broad distribution and a large population size generally maintain high genetic diversity in comparison to species with narrow distribution and small population size.

Heterozygosity can reflect the genetic variation in natural populations and is considered a measure of genetic diversity. The higher the heterozygosity in a population is, the more the genetic variability in it is. *M. officinalis* had high genetic diversity (Ho = 0.513 and He = 0.612) in this study compared to in Luo’s publication for the same species (Ho = 0.3436 and He = 0.2881) [5].

F_ST_ is considered one of the measures of genetic variation in populations. An F_ST_ value of higher than 0.15 can indicate significant differentiation in a population [32]. Therefore, remarkable diversity was observed in the three populations in this study (Table 10). The F_ST_ value ranged from 0.076 (QN population) to 0.451 (TTH population), with an average F_ST_ value of 0.206. This result was higher than that in the study of *P. vietnamensis* (F_ST_ = 0.13) [19] and lower than those in the studies of *Cinnamomum balansae* (F_ST_ = 0.601) [26] and *Pulsatilla patens* (L.) (F_ST_ = 0.22) [33] and (F_ST_ = 0.13) using microsatellite markers.

The results of the AMOVA also revealed that 30% of the total variation was found among subpopulations, while the rest (70%) was within subpopulations. These results indicated a relatively low genetic differentiation among the three populations and a high genetic divergence within those populations. Genetic variation among populations was highly influenced by gene flow, selection, genetic drift, and other factors.

A Bayesian analysis in STRUCTURE showed two different groups of genetically mixed individuals of *M. officinalis*. Even though the populations were separated by large geographical distances, the majority of individuals from the QB and QN populations shared the same ancestral origin.

## 5. Conclusions

The study proved the *ITS1* region and the *ITS2* region as reliable markers in the barcoding of *M. officinalis*. In addition, the results confirmed that *M. officinalis* had low genetic differentiation among the three studied populations and high genetic divergence within those three populations. The three populations of *M. officinalis* were divided into genetic groups on the basis of the codominant genotypic distance. Our study of *M. officinalis* in terms of genetic diversity and population structure is intended to make a large contribution to the conservation of this species.

## Figures and Tables

**Figure 1 genes-13-01938-f001:**
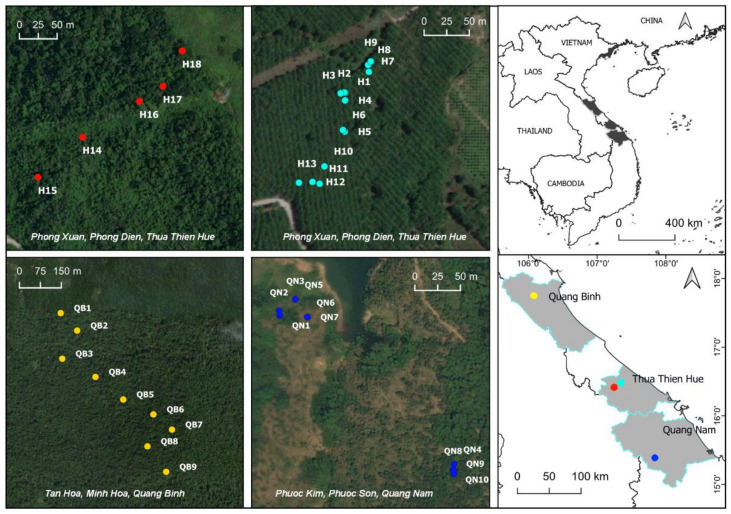
Map showing collection sites of three populations of *M. officinalis* in central Vietnam.

**Figure 2 genes-13-01938-f002:**
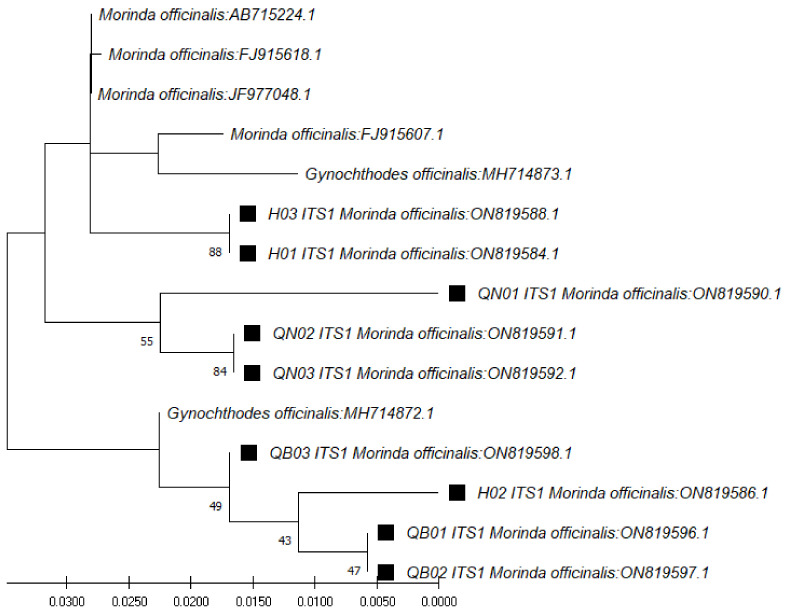
Phylogenetic tree by the maximum likelihood of similarity between *M. officinalis* and the closest species as per *ITS1* region sequences. The samples of our study are marked by black squares next to their names.

**Figure 3 genes-13-01938-f003:**
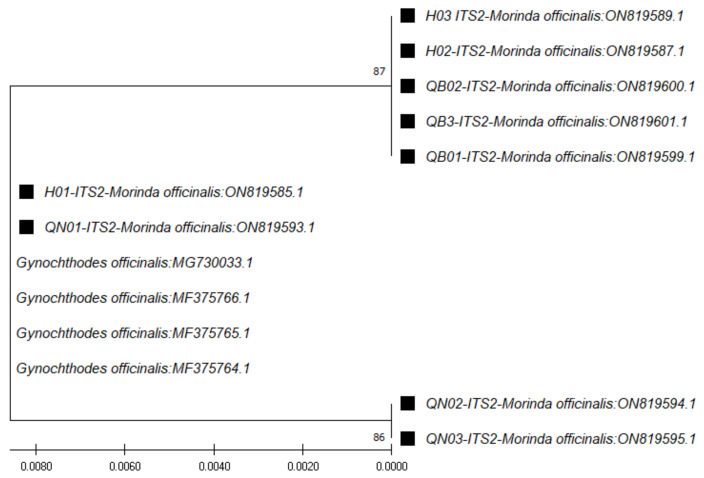
Phylogenetic tree by the maximum likelihood of similarity between *M. officinalis* and the closest species as per *ITS2* region sequences. The samples of our study are marked by black squares next to their names.

**Figure 4 genes-13-01938-f004:**
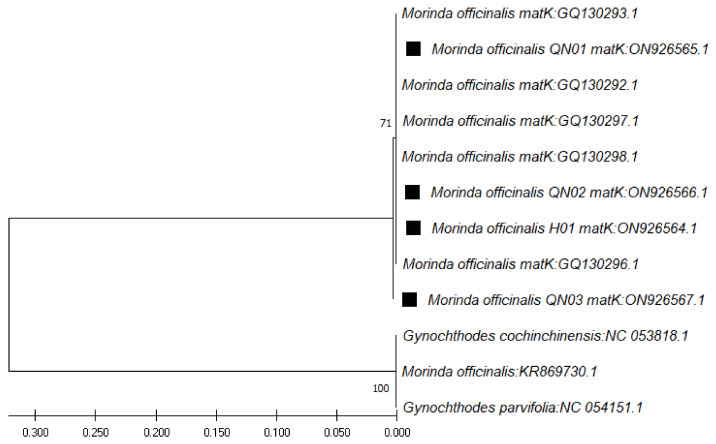
Phylogenetic tree by maximum likelihood of similarity between *M. officinalis* and the closest species as per *matK* region sequences. The samples of our study are marked by black squares next to their names.

**Figure 5 genes-13-01938-f005:**
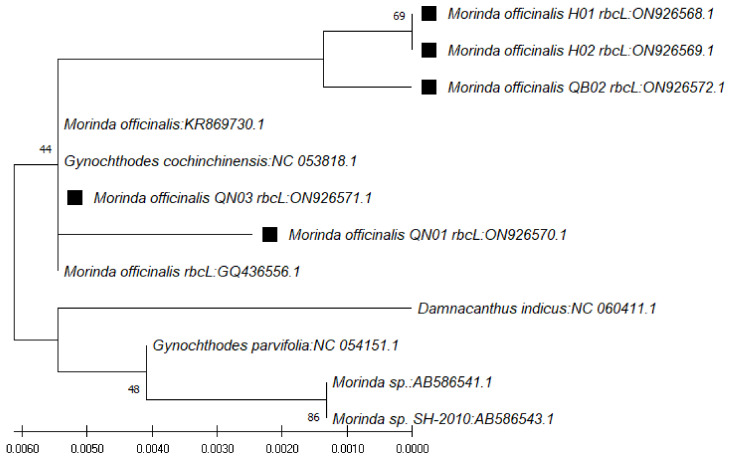
Phylogenetic tree by maximum likelihood of similarity between *M. officinalis* and the closest species as per *rbcL* region sequences. The samples of our study are marked by black squares next to their names.

**Figure 6 genes-13-01938-f006:**
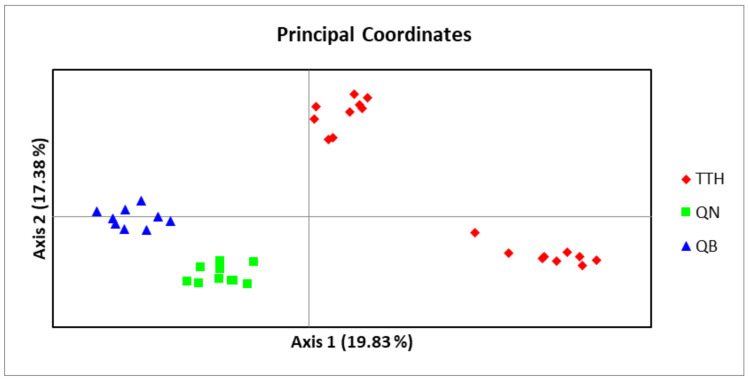
Principal coordinate analysis (PCoA) based on codominant genotypic distance for 37 studied samples of three populations of *M. officinalis (* TTH: TTH population; QN: QN population; QB: QB population).

**Figure 7 genes-13-01938-f007:**
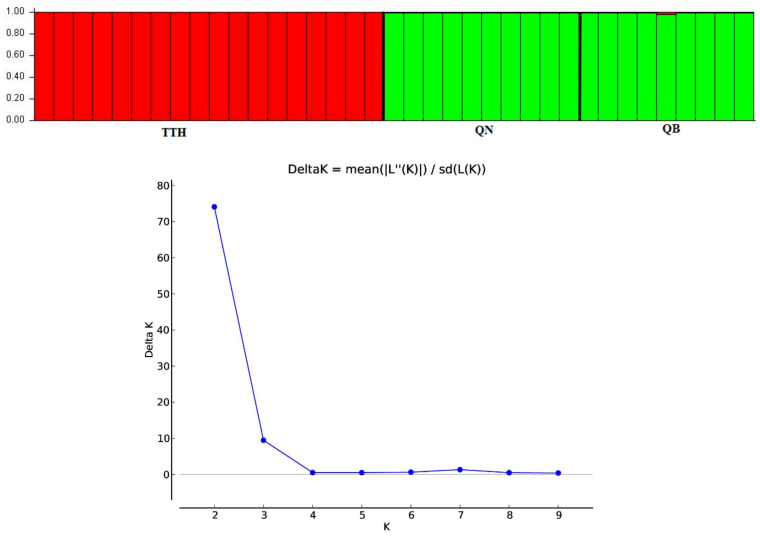
Delta K distribution graph and bar plot of the admixture assignment for three *M. officinalis* populations to clusters (K = 2; highest ΔK value = 74.084117) based on Bayesian analysis.

**Table 1 genes-13-01938-t001:** The universal primers for DNA barcoding used in this study.

Locus	Primer Name	Sequences (5′–3′)	Reference
*ITS1*	*5a fwd*	CCTTATCATTTAGAGCAAGGAG	[12]
	*4 rev*	TCCTCCGCTTATTGATATGC	
*ITS2*	*S2F*	ATGCGATACTTGGTGTGAAT	
	*S3R*	GACGCTTCTCCAGACTACAAT	
*rbcL*	*1f*	ATGTCACCACAAACAGAAAC	
	*724r*	TCGCATGTACCTGCAGTAGC	
*matK*	*390F*	CGATCTATTCATTCAATATTTC	
	*1326R*	TCTAGCACACGAAAGTCGAAGT	

**Table 2 genes-13-01938-t002:** Efficiency of PCR amplification of four potential barcodes in three populations of *M. officinalis*.

Population Codes	Number of Amplified Samples	PCR Efficiency of *ITS1* (%)	PCR Efficiency of *ITS2* (%)	PCR Efficiency of *MatK* (%)	PCR Efficiency of *rbcL* (%)
QB	3	100.0	100.0	0.0	33.3
TTH	3	100.0	100.0	100.0	66.5
QN	3	100.0	100.0	33.3	66.5
Total	9	100	100	44.44	55.56

**Table 3 genes-13-01938-t003:** *M. officinalis* plants in this study and the percentage of similarity to the closest species in the GenBank as per *ITS1* gene sequence.

No.	Species Description	Scientific Name	Aligned *ITS1* Gene Sequence (bp)	Coverage (%) E	Similarity (%)	Accession Numbers
1	H01-*ITS1*-*M.officinalis*	*M. officinalis*	179 (86bp–264bp)	100	100	ON819584.1 ^*^
2	H03-*ITS1*-*M. officinalis*	*M. officinalis*	179 (92bp–270bp)	100	100	ON819588.1 ^*^
3	*Morinda officinalis*	*M. officinalis*	179 (93bp–271bp)	100	98.88	AB715224.1
4	*Morinda officinalis* h15	*M. officinalis*	179 (11bp–189bp)	100	98.88	FJ915618.1
5	*Morinda officinalis* h4	*M. officinalis*	179 (11bp–189bp)	100	98.32	FJ915607.1
6	*Morinda officinalis* voucher	*M. officinalis*	179 (19bp–197bp)	100	97.21	JF977048.1
7	*Gynochthodes officinalis*	*M. officinalis*	179 (73bp–251bp)	100	97.21	MH714873.1
8	QB01-*ITS1*-*M. officinalis*	*M. officinalis*	179 (85bp–263bp)	100	97.21	ON819596.1 ^*^
9	QN03-*ITS1*-*M. officinalis*	*M. officinalis*	179 (83bp–261bp)	100	97.21	ON819592.1^*^
10	QN02-*ITS1*-*M. officinalis*	*M. officinalis*	179 (83bp–261bp)	100	97.21	ON819591.1 ^*^
11	QB03-*ITS1*-*M. officinalis*	*M. officinalis*	179 (86bp–264bp)	100	96.65	ON819598.1 ^*^
12	QB02-*ITS1*-*M. officinalis*	*M. officinalis*	179 (85bp–263bp)	100	96.65	ON819597.1 ^*^
13	H02-*ITS1*-*M. officinalis*	*M. officinalis*	179 (83bp–261bp)	100	95.53	ON819586.1 ^*^
14	*Gynochthodes officinalis*	*M. officinalis*	179 (73bp–251pb)	100	97.21	MH714872.1

Note: “*” is the sample in this study.

**Table 4 genes-13-01938-t004:** *M. officinalis* plants in this study and the percentage of similarity to the closest species in the GenBank as per *ITS2* gene sequences.

No.	Species Description	Scientific Name	Aligned *ITS2* Gene Sequence (bp)	Coverage (%) E	Similarity (%)	Accession Numbers
1	H01-*ITS2*-*M.officinalis*	*M. officinalis*	235 (108bp–342bp)	100	100	ON819585.1 ^*^
2	*Gynochthodes officinalis*	*M. officinalis*	235 (53bp–287bp)	100	100	MG730033.1
3	*Gynochthodes officinalis* B7	*M. officinalis*	235 (102bp–336bp)	100	100	MF375766.1
4	*Gynochthodes officinalis* B5	*M. officinalis*	235 (102bp–336bp)	100	100	MF375765.1
5	*Gynochthodes officinalis* B3	*M. officinalis*	235 (105bp–339bp)	100	100	MF375764.1
6	QB01-*ITS2*-*M.officinalis*	*M. officinalis*	235 (108bp–342bp)	100	99.15	ON819599.1 ^*^
7	QN02-*ITS2*-*M.officinalis*	*M. officinalis*	235 (106bp–340bp)	100	99.15	ON819594.1 ^*^
8	QB03-*ITS2*-*M.officinalis*	*M. officinalis*	235 (108bp–342bp)	100	99.15	ON819601.1 ^*^
9	QB02-*ITS2*-*M.officinalis*	*M. officinalis*	235 (108bp–342bp)	100	99.15	ON819600.1 ^*^
10	QN01-*ITS2*-*M.officinalis*	*G. officinalis*	235 (110bp–344bp)	100	100	ON819593.1 ^*^
11	QN03-*ITS2*-*M.officinalis*	*M. officinalis*	235 (106bp–340bp)	100	99.15	ON819595.1 ^*^
12	H03_*ITS2*-*M.officinalis*	*M. officinalis*	235 (75bp–309bp)	100	99.15	ON819589.1 ^*^
13	H02-*ITS2*-*M.officinalis*	*M. officinalis*	235 (71bp–305bp)	100	99.15	ON819587.1 ^*^

Note: “*” is the sample in this study.

**Table 5 genes-13-01938-t005:** *M. officinalis* plants in this study and the percentage of similarity with the closest species in the GenBank as per *matK* gene sequences.

No.	Species Description	Scientific Name	Aligned Sequence (bp)	Coverage (%) E	Similarity (%)	Accession Numbers
1	*Gynochthodes officinalis* _H01_*matK*	*M. officinalis*	899	100	100	ON926564.1 ^*^
2	*Morinda officinalis*	*M. officinalis*	897	99.00	99.89	KR869730.1
3	*Gynochthodes officinalis*_QN02_*matK*	*M. officinalis*	897	99.00	99.89	ON926566.1 ^*^
4	*Gynochthodes parvifolia*	*G. parvifolia*	897	99.00	99.78	NC_054151.1
5	*Gynochthodes cochinchinensis*	*M. officinalis*	897	99.00	99.78	NC_053818.1
6	*Gynochthodes officinalis_QN03_matK*	*M. officinalis*	897	99.00	99.67	ON926567.1 ^*^
7	*Morinda officinalis* isolate h7_*matK*	*M. officinalis*	864	96.00	99.88	GQ130298.1 ^*^
8	*Morinda officinalis* h2 *_matK*	*M. officinalis*	864	96.00	99.88	GQ130293.1
9	*Morinda officinalis* h1 *_matK*	*M. officinalis*	864	96.00	99.88	GQ130292.1
10	*Morinda officinalis* h6*_matK*	*M. officinalis*	864	96.00	99.77	GQ130297.1
11	*Morinda officinalis* h5	*M. officinalis*	864	96.00	99.77	GQ130296.1
12	*Gynochthodes officinalis*_QN01*_matK*	*M. officinalis*	861	95.00	99.77	ON926565.1 ^*^

Note: “*” is the sample in this study.

**Table 6 genes-13-01938-t006:** *M. officinalis* plants in this study and percentage of similarity with the closest species in the GenBank base as per *rbcL* gene sequences.

No.	Species Description	Scientific Name	Aligned Sequence (bp)	Coverage (%) E	Similarity (%)	Accession Numbers
1	*Gynochthodes officinalis*_H01_*rbcL*	*M. officinalis*	738	100.00	100	ON926568.1 ^*^
2	*Gynochthodes officinalis*_QB02*_rbcL*	*M. officinalis*	738	99.00	99.73	ON926572.1 ^*^
3	*Morinda officinalis*	*M. officinalis*	737	99.00	99.46	KR869730.1
4	*Gynochthodes cochinchinensis*	*M. officinalis*	737	99.00	99.46	NC_053818.1
5	*Gynochthodes parvifolia*	*G. parvifolia*	737	99.00	99.32	NC_054151.1
6	*Gynochthodes officinalis*_H02_*rbcL*	*M. officinalis*	713	96.00	100.00	ON926569.1 ^*^
7	*Morinda* sp.	*Morinda* sp.	737	97.00	99.31	AB586541.1
8	*Morinda* sp.	*Morinda* sp.	721	97.00	99.31	AB586543.1
9	*Morinda officinalis*	*M. officinalis*	703	95.00	99.86	GQ436556.1
10	*Gynochthodes officinalis*_QN03_*rbcL*	*M. officinalis*	670	90.00	99.85	ON926571.1 ^*^
11	*Gynochthodes officinalis*_QN01_ *rbcL*	*M. officinalis*	670	90.00	99.55	ON926570.1 ^*^

Note: “*” is the sample in this study.

**Table 7 genes-13-01938-t007:** Nucleotide components of the *ITS1* and *ITS2* gene regions of nine *M. officinalis* samples.

Sam.	**Compute Nucleotide Composition (%)**											
	** *ITS1* **						*ITS2*					
	T(U)	C	A	G	C + G	Total	T(U)	C	A	G	C + G	Total
H01	12.3	35.8	21.2	30.7	66.5	179	14.5	35.7	15.7	34.0	69.7	235
H02	11.2	36.9	20.1	31.8	68.7	179	14.0	36.2	16.2	33.6	69.8	235
H03	12.3	35.8	21.2	30.7	66.5	179	14.0	36.2	16.2	33.6	69.8	235
QN01	10.0	38.9	20.0	31.1	70	180	14.5	35.7	15.7	34.0	69.3	235
QN02	11.7	35.8	20.7	31.8	66.8	179	14.9	35.3	16.2	33.6	68.9	235
QN03	12.3	36.3	20.1	31.3	67.6	179	14.9	35.3	16.2	33.6	68.9	235
QB1	11.7	36.3	20.7	31.3	67.6	179	14.0	36.2	16.2	33.6	69.8	235
QB2	12.3	36.3	20.1	31.3	67.6	179	14.0	36.2	16.2	33.6	69.8	235
QB3	12.3	36.3	20.1	31.3	67.6	179	14.0	36.2	16.2	33.6	69.8	235
Avg.	11.8	36.5	20.5	31.3	68.0	179.1	14.3	35.9	16.1	33.7	69.6	235

**Table 8 genes-13-01938-t008:** Nucleotide components of the *matk* and *rbcL* gene regions of four *M. officinalis* samples.

Sam.	**Compute Nucleotide Composition (%)**											
	** *matK* **						*rbcL*					
	T(U)	C	A	G	C + G	Total	T(U)	C	A	G	C + G	Total
H01	36.5	17.8	28.8	16.9	34.7	899	28.6	20.3	28.3	22.8	43.1	738
H02	-	-	-	-	-	-	28.1	20.6	28.6	22.7	43.3	713
QN01	36.9	18.2	27.8	17.1	35.3	861	28.5	20.7	27.5	23.3	44.0	670
QN02	36.5	17.8	28.7	17.1	34.9	897	-	-	-	-	-	-
QN03	36.5	18.1	28.4	17.1	35.2	897	28.8	20.7	27.5	23.0	43.7	670
QB2	-	-	-	-	-	-	28.5	20.5	28.2	22.8	43.3	737
Avg.	36.6	18.0	28.4	17.0	35	888.5	28.5	20.6	28.0	22.9	43.5	705.6

**Table 9 genes-13-01938-t009:** Genetic diversity characteristics of 22 microsatellite loci of *M. officinalis*.

Locus	Na	Ne	Ho	He	F_IS_	F_IT_	F_ST_	Nm
MO02	4.667	3.494	0.893	0.698	−0.279	−0.086	0.151	1.405
MO04	4.333	2.902	0.652	0.633	−0.030	0.222	0.244	0.774
MO05	9.000	4.280	0.796	0.733	−0.086	0.109	0.179	1.144
MO12	5.667	3.820	0.815	0.727	−0.121	0.021	0.127	1.718
MO19	7.667	5.277	0.619	0.782	0.209	0.322	0.144	1.491
MO26	6.000	4.061	0.889	0.739	−0.204	−0.015	0.157	1.343
MO30	7.333	5.525	0.963	0.783	−0.230	−0.044	0.152	1.400
MO38	4.000	2.628	0.144	0.574	0.749	0.828	0.318	0.537
MO39	4.000	3.421	0.785	0.669	−0.174	0.071	0.209	0.948
MO41	3.000	2.184	0.689	0.539	−0.278	0.174	0.354	0.457
MO43	2.667	2.142	0.019	0.531	0.965	0.978	0.370	0.425
MO47	3.333	2.091	0.304	0.466	0.349	0.586	0.364	0.437
MO53	3.333	1.892	0.167	0.393	0.576	0.698	0.287	0.621
MO57	7.667	4.660	0.456	0.747	0.391	0.488	0.160	1.311
MO60	5.667	3.501	0.381	0.704	0.458	0.556	0.180	1.139
MO61	5.333	2.863	0.337	0.579	0.418	0.556	0.237	0.805
MO63	4.000	2.333	0.726	0.564	−0.287	0.053	0.265	0.695
MO88	5.000	2.917	0.433	0.652	0.336	0.461	0.189	1.074
MO89	5.333	2.540	0.144	0.425	0.660	0.818	0.463	0.290
MO90	2.333	1.764	0.085	0.396	0.785	0.892	0.498	0.252
MO94	3.333	1.834	0.093	0.405	0.771	0.898	0.554	0.202
MO96	4.667	4.076	0.907	0.736	−0.233	−0.038	0.159	1.327
Mean ± SE					0.216 ± 0.090	0.389 ± 0.077	0.262 ± 0.027	0.900 ± 0.098

**Table 10 genes-13-01938-t010:** Genetic diversity estimates of three populations of *M. officinalis*.

Pop.	N	Na	Np	Ne	P%	Ho	He	F_ST_
QB	9	3.955	1.818	2.567	100.00	0.556	0.554	0.085
TTH	18	6.455	4.364	3.786	100.00	0.394	0.683	0.451
QN	10	4.364	2.318	3.221	95.45	0.591	0.601	0.076
Mean ± SE		4.924 ± 0.296		3.191 ± 0.197	98.48 ± 1.52	0.513 ± 0.047	0.612 ±0.023	0.206 ±0.067

**Table 11 genes-13-01938-t011:** Analysis of molecular variance (AMOVA) for different groups of *M. officinalis* How.

Source	df	SS	MS	Est. Var.	%
Among populations	2	158.697	79.348	3.089	30%
Within populations	71	512.222	7.214	7.214	70%
Total	73	670.919		10.303	100%
Stat	Value	P(rand >= data)			
Fst	0.300	0.001			

## Data Availability

The datasets generated during and/or analyzed during the current study are available from the corresponding author on reasonable request.

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
