# Peer review of "Phylogenetic Analysis Based on DNA Barcoding and Genetic Diversity Assessment of Morinda officinalis How in Vietnam Inferred by Microsatellites"

_genes, 2022, doi:10.3390/genes13111938_

Round 1
Reviewer 1 Report
The research topic is interesting and significant in terms of the biology of the species. The research experiment is designed correctly, and the obtained results are appropriate. However, the presentation of these results could be better by clarifying the figures and tables (indicated in the comments in the manuscript) and by describing them. Improvements are also needed in the discussion part. Please see the attached manuscript file for specific recommendations. The most significant improvements (in principle should be completely reworked) are necessary for the conclusions. At this point, it's a general statement that doesn't really relate to the rest of the text. Conclusions should be specific, arising from the results and discussion.

Author Response
- Materials and Methods
It would be good to add a map with sample collection locations. This would show how far apart these places are, thus how great their potential relationship is.
Authors: Because we need more time to make the map, So the map will be added after the spelling correction service is completed.
- Line 77: the name of dye?
Authors: “Safe dye” is the name of the dye (according to manufacturer).
3. How is this similarity calculated? According to the table, it appears that the samples analyzed in the study have been placed in the gene bank - what have they been compared with? (Table 3, Table 4, Table 5, Table 6).
Authors: First, we took a sample of our research (e.x: Accession number: ON819584.1), and then we used Blast to compare this sample with the rest of the samples and the samples of other research (randomly) in the gene bank. Based on the results on Blast, we will take the samples with the highest similarity rate to the standard sample (e.x: Accession number: ON819584.1) that we used initially for comparison.
- The figure needs an explanation of what the filled squares next to the names of the samples mean.
Authors: The samples in our study are marked with black squares next to the name of the samples. We added the explanation in the manuscript.
Note: The highlights that the reviewer made have been edited.
Reviewer 2 Report
The work is well presented but there are many mistakes which must be addressed.
Line first of abstract very poor sentence and not clarified. It must be revised.
Add future perspective in the abstract.
Introduction is very short and specific.
Add significance of the study in the introduction.
Significance and advances of the phylogenetic studies must be added. The following studies could be cited. https://doi.org/10.1111/jse.12642, Pak. J. Bot., 54(3): DOI: http://dx.doi.org/10.30848/PJB2022-3(19),
Medicinal and economic importance of the study is also required.
Current advances of the DNA barcoding must be provided.
Add reference in the DNA extraction https://doi.org/10.1016/j.jplph.2019.152997,
Line 135 Bayesian analysis provide link of the software.
English of the whole MS must be improved
Conclusion is well justified. The authors should discuss some points for the future studies molecular level studies and DNA barcoding.
Author Response
- Line first of abstract very poor sentence and not clarified. It must be revised.
Authors: 1st line of abstract was revised. The revised sentence below:
“Morinda officinalis How. is well known as one of the most valuable medicinal plants in some regions of Vietnam. This species is mainly used for treating male impotence, irregular menstruation, and rheumatoid arthritis”
- Add future perspective in the abstract.
Authors: The future perspective was added in the abstract.
- Introduction section: Introduction is very short and specific. Add significance of the study in the introduction. Significance and advances of the phylogenetic studies must be added. The following studies could be cited. https://doi.org/10.1111/jse.12642, Pak. J. Bot., 54(3): DOI: http://dx.doi.org/10.30848/PJB2022-3(19),. Medicinal and economic importance of the study is also required. Current advances of the DNA barcoding must be provided.
Authors: Introduction section was revised based on the author’s comments.
- Line 135 Bayesian analysis provide link of the software.:
Authors: we added the link of software (https://web.stanford.edu/group/pritchardlab/structure.html) in line 135
- English of the whole MS must be improved
Authors: We used the editing services listed at https://www.mdpi.com/authors/english
- Conclusion is well justified. The authors should discuss some points for the future studies molecular level studies and DNA barcoding.
Authors: Conclusion was revised.
Note: All revisions to the manuscript were marked up using the “Highlight” function

Round 2
Reviewer 1 Report
The manuscript has been improved according to the suggestions given. It can be published in the existing version.
The only thing that could be improved is the caption of Figure 6: it should indicate what TTH, QN and QB mean. The image caption should be informative regardless of the text.